# The Spike D614G mutation increases SARS-CoV-2 infection of multiple human cell types

Zharko Daniloski[1,2†], Tristan X Jordan[3†], Juliana K Ilmain[4], Xinyi Guo[1,2], Gira Bhabha[4], Benjamin R tenOever[3*], Neville E Sanjana[1,2*]

[1]New York Genome Center, New York, United States; [2]Department of Biology, New York University, New York, United States; [3]Department of Microbiology, Icahn School of Medicine at Mount Sinai, New York, United States; [4]Department of Cell Biology and Skirball Institute of Biomolecular Medicine, New York University School of Medicine, New York, United States

**Abstract** A novel variant of the SARS-CoV-2 virus carrying a point mutation in the Spike protein (D614G) has recently emerged and rapidly surpassed others in prevalence. This mutation is in linkage disequilibrium with an ORF1b protein variant (P314L), making it difficult to discern the functional significance of the Spike D614G mutation from population genetics alone. Here, we perform site-directed mutagenesis on wild-type human-codon-optimized Spike to introduce the D614G variant. Using multiple human cell lines, including human lung epithelial cells, we found that the lentiviral particles pseudotyped with Spike D614G are more effective at transducing cells than ones pseudotyped with wild-type Spike. The increased transduction with Spike D614G ranged from 1.3- to 2.4-fold in Caco-2 and Calu-3 cells expressing endogenous ACE2 and from 1.5- to 7.7-fold in A549[ACE2] and Huh7.5[ACE2] overexpressing ACE2. Furthermore, *trans*-complementation of SARS-CoV-2 virus with Spike D614G showed an increased infectivity in human cells. Although there is minimal difference in ACE2 receptor binding between the D614 and G614 Spike variants, the G614 variant is more resistant to proteolytic cleavage, suggesting a possible mechanism for the increased transduction.

**\*For correspondence:**
benjamin.tenoever@mssm.edu
(BRO);
neville@sanjanalab.org (NES)

[†]These authors contributed equally to this work

## Introduction

Recently, a novel variant of the SARS-CoV-2 virus carrying a point mutation in the Spike protein (D614G) has emerged and rapidly surpassed others in prevalence, including the original SARS-CoV-2 isolate from Wuhan, China. This Spike variant is a defining feature of the most prevalent clade (A2a) of SARS-CoV-2 genomes worldwide (*Bhattacharyya et al., 2020*; *Hadfield et al., 2018*). Using phylogenomic data, several groups have proposed that the D614G variant may confer increased transmissibility leading to positive selection (*Bhattacharyya et al., 2020*), while others have claimed that currently available evidence does not support positive selection (*Dorp et al., 2020*; *Korber et al., 2020*). Furthermore, in the A2a clade, this mutation is in linkage disequilibrium with a ORF1b protein variant (P314L) (*Bhattacharyya et al., 2020*), making it difficult to discern the functional significance of the Spike D614G mutation from population genetics alone.

Here, we perform site-directed mutagenesis on a human-codon-optimized Spike protein to introduce the D614G variant (*Shang et al., 2020*) and produce SARS-CoV-2-pseudotyped lentiviral particles (S-virus) with this variant and with D614 Spike. We show that in multiple cell lines, including human lung epithelial cells, that S-virus carrying the D614G mutation is up to eightfold more effective at transducing cells than wild-type S-virus. Similar experiments using intact SARS-CoV-2 further confirm that Spike G614 leads to higher viral infection of human cells. Although we find minimal

differences in ACE2 receptor binding between the Spike variants, we show that the G614 variant is more resistant to cleavage in human cells, which may suggest a possible mechanism for the increased transduction. Given that several vaccines in development and in clinical trials are based on the initial (D614) Spike sequence (*Lurie et al., 2020*; *Yu et al., 2020*), this result has important implications for the efficacy of these vaccines in protecting against this recent and highly prevalent SARS-CoV-2 variant. For example, neutralizing antibodies that target the receptor-binding domain seem largely unaffected in potency, but it remains to be seen whether the D614G variant alters neutralization sensitivity to other classes of anti-Spike antibodies (*Yurkovetskiy et al., 2020*).

## Results

The first sequenced SARS-CoV-2 isolate (GenBank accession MN908947.3) and the majority of viral sequences acquired in January and February 2020 contained an aspartic acid at position 614 of the Spike protein (*Figure 1a*). Beginning in February 2020, an increasing number of SARS-CoV-2 variants with glycine at position 614 of the Spike protein were identified. We found that ~72% of 22,103 SARS-CoV-2 genomes that we surveyed from the GISAID public repository in early June 2020 contained the G614 variant (*Shu and McCauley, 2017*). Previously, Cardozo and colleagues reported a correlation between the prevalence of the G614 variant and the case-fatality rate in individual localities using viral genomes available through early April 2020 (*Becerra-Flores and Cardozo, 2020*). Using a ~10-fold larger dataset, we found a smaller yet significant positive correlation between the prevalence of G614 in a country with its case-fatality rate ($r_p$ = 0.29, *p*=0.04) (*Figure 1b*). There has been little consensus on the potential function of this mutation and whether its spread may or may not be due to a founder effect (*Bhattacharyya et al., 2020*; *Dorp et al., 2020*). Recently, two separate groups at the University of Sheffield and at the University of Washington have found that in COVID-19 patients there is a ~3-fold increase in viral RNA during quantitative PCR (qPCR) testing for those patients with the G614 variant (*Korber et al., 2020*; *Wagner et al., 2020*; *Figure 1c,d*). Using data from these published studies (*Korber et al., 2020*; *Wagner et al., 2020*), we found a consistent difference in qPCR amplification (~5 $C_t$) potentially due to different sampling procedures, RNA extraction and reverse transcription methods, qPCR reagents, or threshold cycle settings (*Figure 1c*). However, the difference in amplification ($\Delta\Delta C_t$) between G614 and D614 variants is remarkably consistent (1.6 $C_t$ for Sheffield, 1.8 $C_t$ for Washington), suggesting that this may be due to a biological difference between COVID-19 patients with specific Spike variants (*Figure 1d*).

Given these findings, we wondered whether the G614 variant may confer some functional difference that impacts viral transmission or disease severity. To address this question, we used a pseudo-typed lentiviral system similar to those developed previously for SARS-CoV-1 (*Moore et al., 2004*). Using site-directed mutagenesis and a full-length human-codon-optimized SARS-CoV-2 Spike coding sequence (*Shang et al., 2020*), we constructed EGFP-expressing lentiviruses either lacking an attachment protein or pseudotyped with D614 Spike or G614 Spike (*Figure 2a*). After production and purification of these viral particles, we transduced four human cell lines derived from lung (A549, Calu-3), liver (Huh7.5), and colon (Caco-2). Caco-2 and Calu-3 cells express endogenous angiotensin-converting enzyme 2 (ACE2) (*Ren et al., 2006*); for A549 and Huh7.5, we overexpressed ACE2 using a lentiviral construct (A549[ACE2], Huh7.5[ACE2]). Others have observed increased S-virus transduction in cells that overexpress the ACE2 receptor (*Li et al., 2003*; *Moore et al., 2004*); we also found that S-virus is much more efficient at transducing human cell lines when the human ACE2 receptor is overexpressed (*Figure 2—figure supplement 1a*).

After transduction with four different viral volumes, we waited 3 days and then performed flow cytometry to measure GFP expression (*Figure 2b*, *Figure 2—figure supplement 1b*). We found in all four human cell lines at all viral doses that G614 S-virus resulted in a greater number of transduced cells than D614 S-virus (*Figure 2c*). Lentivirus lacking an attachment protein resulted in negligible transduction (*Figure 2c*). With the G614 Spike variant, the increase in viral transduction over the D614 variant ranged from 1.4- to 1.9-fold for Calu-3, 1.3- to 2.4-fold for Caco-2 colon, 1.8- to 4.6-fold for A549[ACE2] lung, and 1.5- to 7.7-fold for Huh7.5[ACE2] liver, depending on the specific dose of S-virus (*Figure 2d*). To control for any potential differences in viral titer, we also measured viral RNA content by qPCR. We observed only a small difference between D614- and G614-pseudotyped viruses using two independent primer sets (average of 7% higher viral titer for D614), which may



**Figure 1.** The SARS-CoV-2 D614G mutation has spread rapidly and is correlated with increased fatality and higher viral load. (a) Prevalence of D614G-containing SARS-CoV-2 genomes over time. This visualization was produced by the Nextstrain webtool using GISAID genomes (n = 3,866 genomes samples from January 2020 to May 2020). (b) Per-country Pearson correlation of G614 prevalence with the case-fatality rate (n = 56 countries and 22,103 genomes). (c) Threshold cycle for quantitative polymerase chain reaction (qPCR) detection of SARS-CoV-2 from patients with D614 and G614 Spike. Numbers in parentheses indicate the number of COVID-19 patients in each group and significance testing is using the Wilcoxon rank sum test. This Sheffield data was originally presented in *Korber et al., 2020*. The University of Washington data was originally presented in *Wagner et al., 2020*. (d) Fold-change of increase in viral RNA present in COVID-19 patient samples with G614 Spike as compared to those with D614 Spike.

result in a slight underestimation of the increase in transduction efficacy of the G614-pseudotyped virus (*Figure 2—figure supplement 2*).

We next sought to understand the mechanism through which the G614 variant increases viral transduction of human cells. Like SARS-CoV-1, the SARS-CoV-2 Spike protein has both a receptor-binding domain and a hydrophobic fusion polypeptide that is used after binding the receptor (e.g. ACE2) to fuse the viral and host cell membranes (*Heald-Sargent and Gallagher, 2012*; *Figure 3a*). Initially, we hypothesized that the increased viral transduction of the G614 variant may be due to enhanced binding of the ACE2 receptor. To determine whether greater transduction efficiency results from increased affinity of Spike G614 to its receptor, we used bio-layer interferometry to measure the binding kinetics of the Spike protein S1 subunit, which contains the ACE2-binding site, with and without the variant. We observed similar binding profiles of soluble D614 and G614 Spike S1 subunit to immobilized hACE2 (*Figure 3b,c*). Binding was best represented by a 2:1

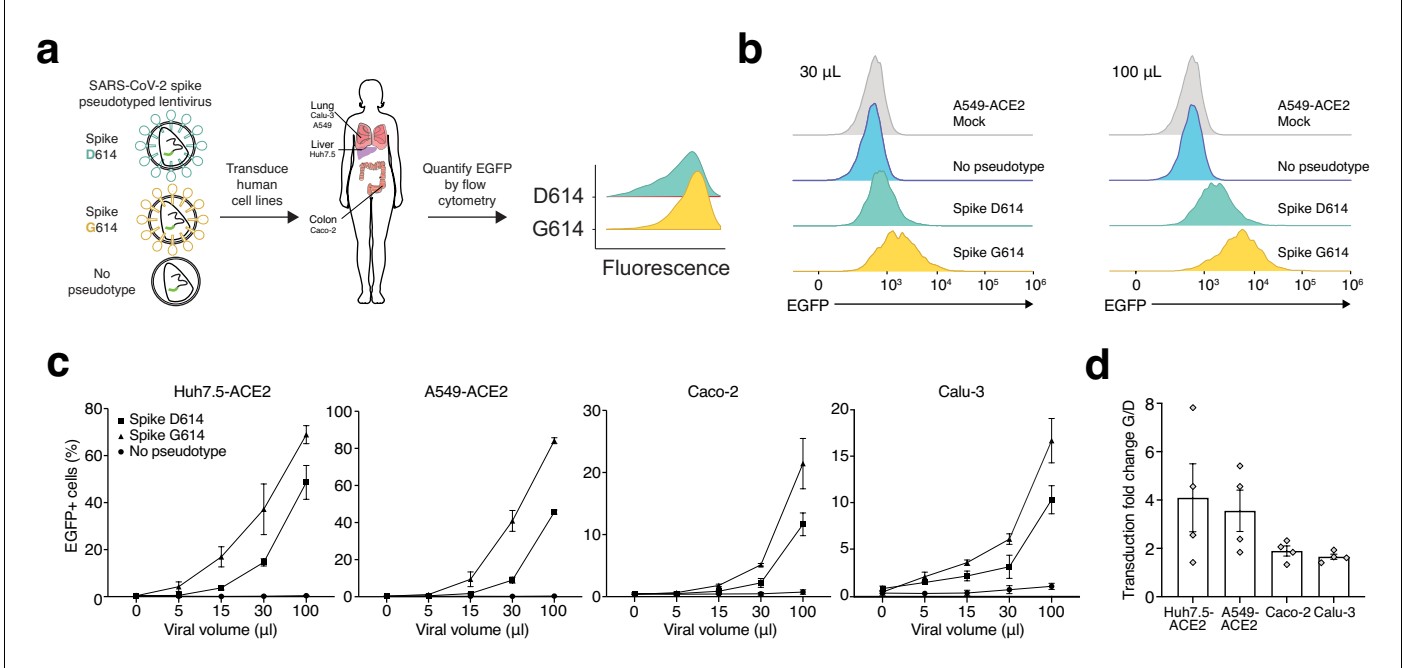

**Figure 2.** SARS-CoV-2 Spike D614G-pseudotyped lentivirus results in increased transduction of human lung, liver, and colon cell lines. (a) Schematic of EGFP lentivirus pseudotyped with SARS-CoV-2 Spike proteins (or no pseudotype) and quantification of EGFP fluorescence by flow cytometry. (b) Flow cytometry of A549[ACE2] cells at 3 days post-transduction with 30 or 100 μL SARS-CoV-2 Spike-pseudotyped lentivirus. (c) Percent of EGFP+ cells at 3 days post-transduction with the indicated volume of virus and pseudotype in human liver Huh7.5[ACE2] cells, lung A549[ACE2] cells, colon Caco-2 cells, and lung Calu-3 cells (n = 3 replicates, error bars are s.e.m.). (d) The fold-change for each viral transduction of G614 Spike over D614 Spike in four cell lines (error bars are s.e.m.).

The online version of this article includes the following figure supplement(s) for figure 2:

**Figure supplement 1.** Increased transduction of SARS-CoV-2 Spike-pseudotyped lentivirus in human cells that constitutively overexpress the human ACE2 receptor.

**Figure supplement 2.** Quantitative PCR of viral RNA from SARS-CoV-2 Spike-pseudotyped lentiviruses.

heterogeneous binding model, which reports two dissociation constants ($K_D$): $K_{D1}$ = 8.45 nM and $K_{D2}$ = 127 nM for D614 variant, and $K_{D1}$ = 18.0 nM and $K_{D2}$ = 92.7 nM for G614 variant (*Table 1*). $K_{D1}$ is consistent with previously published binding affinities of the Spike D614–ACE2 interaction (*Walls et al., 2020*; *Yi et al., 2020*) and is similar for the Spike G614 variant. This suggests our observed transduction phenotype is independent of Spike S1 subunit affinity for the ACE2 receptor.

In order for SARS-CoV-2 to enter cells, the Spike protein must be cleaved at two sites by host proteases. It is thought that Spike must first be cleaved into S1 and S2 fragments, which exposes another cleavage site (*Bestle et al., 2020*; *Hoffmann et al., 2020*). The second cleavage event (creating the S2' fragment) is thought to enable membrane fusion with the host cell. We transfected both D614 and G614 Spike variants into human HEK293FT cells to see whether Spike cleavage might differ between these variants. Both constructs were tagged at their C-termini with a C9 tag to visualize full-length, S2, and S2' fragments via western blot (*Figure 3d*). To measure cleavage, we quantified the ratio of cleaved Spike (S2 + S2') to full-length Spike (*Figure 3e*). We found that the G614 variant is ~2.5 fold more resistant to cleavage in the host cell than the D614 variant in cells expressing Spike by transient transfection (*Figure 3f*). Next, we measured the ratio of cleaved to full-length Spike on Spike-pseudotyped lentiviral particles (*Figure 3g,h*). We found that virions with the G614 variant have less Spike cleavage compared to the D614 variant (~1.4-fold) (*Figure 3i*). This suggests that the increased transduction observed with G614 S-virus (*Figure 2d*) may be due to the greater fraction of uncleaved G614 Spike, which perhaps enable each newly-assembled virion to include more receptor-binding-capable Spike protein. Lastly, we quantified the total Spike protein (cleaved and uncleaved) in each S-virus and normalized it to the pseudovirus capsid protein p24 (*Figure 3—*

**a**

SARS-CoV-2 Spike protein

N-term domain | Receptor binding domain | D614G | Fusion polypeptide | Trans-membrane | C9-tag

S1 | S2

S2'

**b** D614

Shift (nm)

0 100 200 300 400

**c** G614

Shift (nm)

Time (s)

0 100 200 300 400

**d** HEK293 tx:

Mock | D614 | G614 | Spike

180 — ◄ S
130 —
100 — ◄ S2
       ◄ S2'
40 —
35 — ◄ GAPDH

**e**

Spike cleaved to uncleaved (%)

***

tx Spike: D614 G614

**f**

Cleavage ratio (D614 / G614)

Cleaved spike

**g** boiled S-virus:

D614 | G614 | Spike

180 — ◄ S
130 —
100 — ◄ S2
       ◄ S2'
25 — ◄ p24

**h**

Spike cleaved to uncleaved (%)

*

S-virus Spike: D614 G614

**i**

Cleavage ratio (D614 / G614)

Cleaved spike

**Figure 3.** The SARS-CoV-2 Spike D614G variant displays similar ACE2 binding kinetics but altered proteolytic cleavage. (**a**) Schematic diagram of SARS-CoV-2 Spike protein structure with the added C9 affinity tag on the C-terminus. Spike cleavage fragments S1, S2, and S2' are also indicated. (**b, c**) Association and dissociation binding curves of Spike D614 (**b**) and Spike G614 (**c**) with hACE2. Blue curves represent Spike protein binding

*Figure 3 continued on next page*

*Figure 3 continued*

profiles at 200 nM, 100 nM, 50 nM, 25 nM, and 6.25 nM. Red curves represent the best global fit using a 2:1 heterogeneous ligand model. (**d**) Western blot of total protein lysate from HEK293FT cells after transfection with D614 Spike, G614 Spike, or mock transfection. (Upper) Detection of full-length Spike and cleavage fragments using an anti-C9 (rhodopsin) antibody. (Lower) Detection of GAPDH via anti-GAPDH antibody. (**e**) Fraction of cleaved (S2 + S2') to uncleaved (full-length) fragments for Spike D614 and G614 (n = 4 replicates, error bars are s. e.m.). (**f**) Fold-change in cleavage between Spike variants (D614/G614) (n = 4 replicates, error bars are s.e.m.). (**g**) Western blot of boiled S-virus pseudotyped with D614 Spike or G614 Spike. (Upper) Detection of full-length Spike and cleavage fragments using an anti-C9 (rhodopsin) antibody. (Lower) Detection of p24 capsid protein via anti-24 antibody. (**h**) Fraction of cleaved (S2 + S2') to uncleaved (full-length) fragments for Spike D614 and G614 on lentiviral particles (n = 4 replicates, error bars are s.e.m.). (**i**) Fold-change in cleavage between Spike variants on lentiviral particles (D614/G614) (n = 4 replicates, error bars are s.e.m.). *p≤0.05, ***p≤0.001. Significance testing was done with an unpaired two-tailed *t*-test.

The online version of this article includes the following figure supplement(s) for figure 3:

**Figure supplement 1.** Spike incorporation into Spike-pseudotyped lentiviral particles.

**Figure supplement 2.** Change in MHC binding affinity for peptides near the D614G mutation in the SARS-CoV-2 Spike protein.

---

*figure supplement 1*). We found no significant difference between Spike D614 and Spike G614 incorporation into pseudotyped lentiviral particles.

Given the global efforts to develop a COVID-19 vaccine, we also sought to understand the impact of the Spike variant on immune responses. According to the World Health Organization, there are presently 174 COVID-19 vaccine candidates in preclinical development and 63 vaccine candidates in patient-enrolling clinical trials (https://www.who.int/publications/m/item/draft-landscape-of-covid-19-candidate-vaccines, accessed on January 26, 2021). Despite the tremendous diversity of vaccine formulations and delivery methods, many of them utilize Spike sequences (RNA or DNA) or peptides and were developed prior to the emergence of the G614 variant. Using epitope prediction for common HLA alleles (*Andreatta and Nielsen, 2016*), we found that the G614 variant can alter predicted MHC binding ( Figure 3-figure supplement 2). For example, the predicted binding for one high-affinity epitope decreased by nearly fourfold (58 nM for D614 versus 221 nM for G614 with HLA-A*02:01). Although full-length Spike protein likely can produce many immunogenic peptides, several vaccines use only portions of Spike (*Lurie et al., 2020*; *Yu et al., 2020*) and thus it may be better to adapt vaccine efforts to the D614 variant given its global spread.

To understand whether the observed change in transduction efficiency that we found with our pseudotyped lentivirus also impacts full SARS-CoV-2 virus, we sought to develop an isogenic system for testing the Spike variant. Isolates from patient samples that carry the Spike variant also carry a linked mutation in ORF1b, which makes it challenging to perform this experiment in an isogenic fashion. In lieu of a reverse genetics system to generate a SARS-CoV-2 variant and building on the observation by several groups that most cell lines require ACE2 overexpression for efficient SARS-CoV-2 infection (*Hoffmann et al., 2020*; *Ou et al., 2020*; *Shang et al., 2020*; *Ziegler et al., 2020*), we developed a novel *trans*-complementation assay in which we co-transfect either D614 or G614 Spike along with human ACE2 into HEK293T cells (*Figure 4a*). Twenty-four hours later, these cells were infected with SARS-CoV-2 at a low multiplicity of infection (MOI): In this manner, only transfected cells, which express ACE2 (and one of the Spike variants), can be readily infected by SARS-CoV-2.

---

**Table 1.** SARS-CoV-2 Spike D614 and G614 variants have comparable binding affinities for hACE2. Values of kinetics assay from bio-layer interferometry with purified Spike and ACE2. A 2:1 model yields two $K_D$ values as measured by $K_{dis}/K_{on}$ for each binding process.

| | $K_D1$ (nM) | $K_{on}1$ (1/M * s) | $K_{dis}1$ (1/s) | $K_D2$ (nM) | $K_{on}2$ (1/M * s) | $K_{dis}2$ (1/s) |
|---|---|---|---|---|---|---|
| Spike D614 | 8.45 | $3.51 \times 10^4$ | $2.96 \times 10^{-4}$ | 127 | $1.59 \times 10^5$ | $2.01 \times 10^{-2}$ |
| Spike G614 | 18.0 | $4.25 \times 10^4$ | $7.67 \times 10^{-4}$ | 92.7 | $2.22 \times 10^5$ | $2.06 \times 10^{-2}$ |

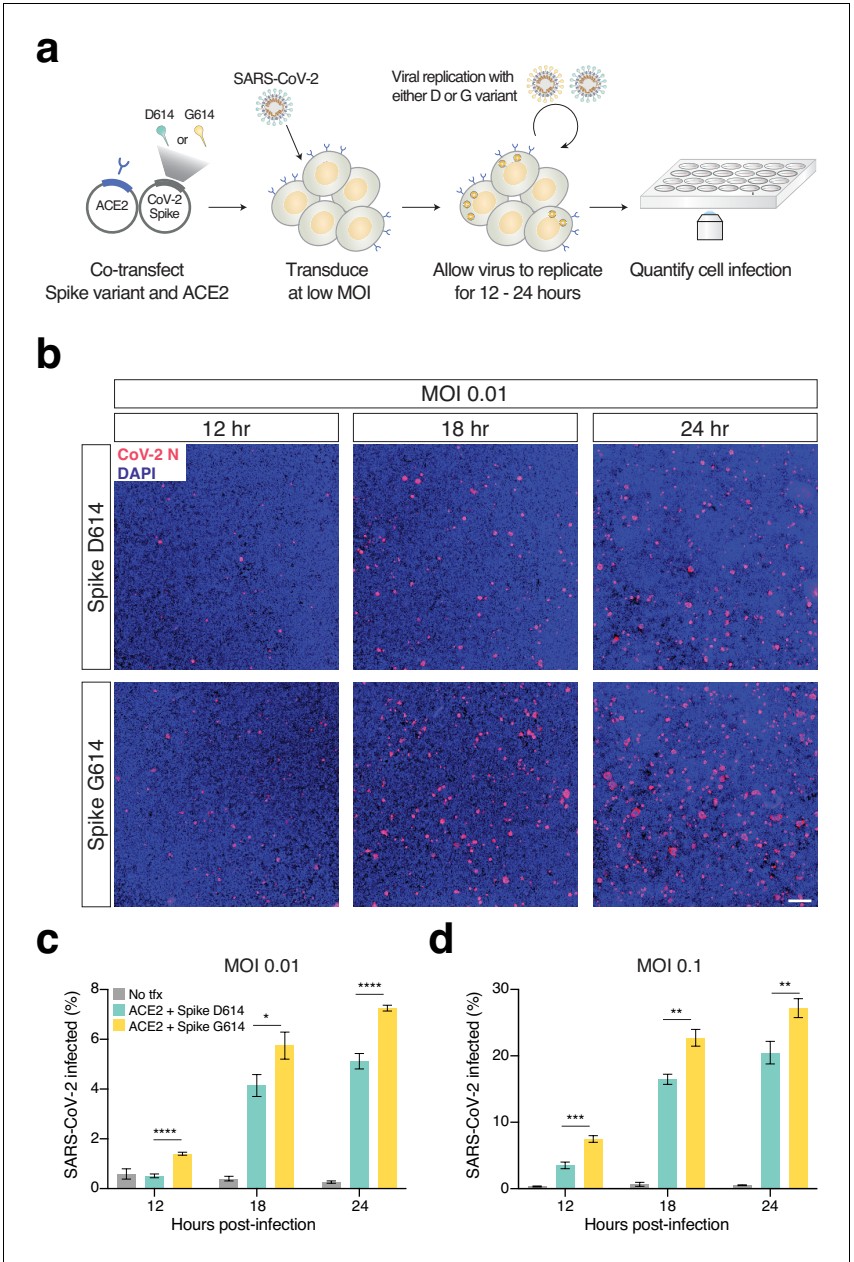

**Figure 4.** Increased infection of SARS-CoV-2 virus with the Spike D614G variant in human cells. (**a**) Schematic diagram of *trans*-complementation assay to assess the impact of SARS-CoV-2 Spike variants in an isogenic fashion. Plasmids containing a single Spike variant and ACE2 are co-transfected into HEK293T cells and then, after 24 hr, are infected with SARS-CoV-2 (Isolate USA-WA1/2020). (**b**) Representative images of HEK293T cells infected with SARS-CoV-2 (multiplicity of infection [MOI]: 0.01) and fixed at the indicated time point post-infection. The cells were stained with DAPI (blue) and an antibody for SARS-CoV-2 nucleocapsid protein (red). (Top) Transfection of ACE2 and SARS-CoV-2 Spike D614 plasmids. (Bottom) Transfection of ACE2 and SARS-CoV-2 Spike G614 plasmids. Scale bar: 1 mm. (**c, d**) Percent of SARS-CoV-2-infected cells at 12, 18, or 24 hr post-infection as measured by immunocytochemistry for SARS-CoV-2 nucleocapsid (N) protein and quantification using an imaging cytometer. A MOI of 0.01 is in shown in (**c**) and a MOI of 0.1 is shown in (**d**). *p≤0.05, **p≤0.01, ***p≤0.001, ****p≤0.0001. Significance testing was done with an unpaired two-tailed t-test.

The online version of this article includes the following figure supplement(s) for figure 4:

**Figure supplement 1.** *Trans*-complementation of full SARS-CoV-2 virus with Spike G614 increases viral infection.

We performed this experiment at two different MOIs (0.01 and 0.1) and measured differences in viral infection at 12, 18, and 24 hr post-infection (*Figure 4b*, *Figure 4—figure supplement 1*), as higher MOIs and longer infection periods may mask the contribution of transfected Spike in this assay. We found significant increases in infection with SARS-CoV-2 complemented with Spike G614 at all time points and all MOIs (*Figure 4c,d*). As a negative control, infected HEK293T cells without any prior transfection resulted in minimal SARS-CoV-2 infection (<1% in most cases). Using the *trans*-complementation assay, we have shown that introduction of just the Spike D614G point mutation increases infection using intact (replication-competent) SARS-CoV-2 virus.

## Discussion

In summary, we have demonstrated that the recent and now-dominant mutation in the SARS-CoV-2 Spike glycoprotein D614G increases the efficiency of cellular entry for the virus across a broad range of human cell types, including cells from lung, liver, and colon. We showed increased entry efficiency using both a pseudotyped lentiviral model system and a replication-competent SARS-CoV-2 virus. Given the concordance between the pseudotyped lentiviral system and SARS-CoV-2 virus, this suggests that changes in Spike protein are well represented using the pseudovirus, which should enable a much broader group of laboratories to use and study Spike variants.

We also found that G614 Spike is more resistant to proteolytic cleavage during production of the protein in host cells as well as on pseudotyped lentiviral particles, suggesting that replicated virus produced in human cells may be more infectious due to a greater proportion of functional (uncleaved with receptor-binding domain) Spike protein per virion. In contrast to a recent study with Spike-pseudotyped murine leukemia virus (*Zhang et al., 2020*), we did not observe a difference in Spike incorporation into lentiviral particles with the G614 variant as compared to the D614 variant, which is consistent with observations by other groups (*Ozono et al., 2020*; *Yurkovetskiy et al., 2020*). It is now well established that SARS-CoV-2 Spike is proteolytically processed primarily by furin; however, new reports suggest that G614 maybe also processed by elastase-2 (*Hu et al., 2020*). Future research is required to confirm these findings and establish if Spike G614 is proteolytically processed differently on live SARS-CoV-2 virus. Using bio-layer interferometry with purified SARS-CoV-2 Spike S1 fragment monomers and ACE2 proteins, we found no significant difference in binding kinetics with the ACE2 receptor resulting from the D614G mutation. Yurkovetskiy and colleagues reported that, with purified Spike full-length trimers, the G614 mutation shifts the Spike protein conformation to an ACE2-binding competent state (*Yurkovetskiy et al., 2020*), suggesting that Spike full-length trimers behave differently than the Spike S1 monomers that we investigated in this study.

Several other groups have also reported that the D614G results in increased viral fitness and infection efficiency (*Hu et al., 2020*; *Jiang et al., 2020*; *Li et al., 2020*; *Ozono et al., 2020*; *Plante et al., 2020*; *Yurkovetskiy et al., 2020*; *Zhang et al., 2020*). Most of them utilized Spike-pseudotyped viral particles to demonstrate this finding. While the common consensus is that Spike G614 mutation increases viral fitness and infectivity, the mechanism by which it occurs varies among these studies. Several of these studies investigated the proteolytic processing, incorporation, and ACE2-binding conformation state of the Spike protein, with some discrepancies between them regarding the proteolytic processing and incorporation of Spike. This can be explained, at least in part, by the technical differences in the protocols used to generate the Spike-pseudotyped viral particles. For example, the studies use a wide variety of pseudoviral systems, such as murine leukemia virus (Zhang et al., 2020), mouse sarcoma virus (Jiang et al., 2020), and lentivirus (our study and Yurkovetskiy et al., 2020). While all four studies utilized HEK293 cells to generate the particles, each one used different transfection methods and it is possible that the transfection reagents can adversely affect the cell state and protease activity. Two studies that utilized isogenic SARS-CoV-2 D614 and G614 variants found no difference in Spike protein cleavage and incorporation into viral particles (*Hou et al., 2020*; *Plante et al., 2020*). The differences in Spike biology between isogenic pseudoviruses and SARS-CoV-2 viruses may be due to differences in Spike protein trimer assembly and presentation, as well as the absence/presence of additional SARS-CoV-2 proteins. Altogether, these differences illustrate the advantages and limitations of using pseudoviruses and isogenic viruses that should be taken into consideration in future studies.

Despite the emerging consensus the G614 results in faster viral spread (*Korber et al., 2020*), it is still uncertain whether this will have a clinical impact on COVID-19 disease progression. Two studies that have examined potential differences in clinical severity or hospitalization rates did not see a correlation with Spike mutation status (*Korber et al., 2020*; *Wagner et al., 2020*), although one study found a small but not significant enrichment of G614 mutations among intensive care unit patients (*Korber et al., 2020*). Given its rapid rise in human isolates and enhanced transduction across a broad spectrum of human cell types, the G614 variant and other recent variants merit careful consideration by biomedical researchers working on candidate therapies, such as those to modulate cellular proteases, and on vaccines that deliver Spike D614 nucleic acids or peptides.

# Materials and methods

## Key resources table

| Reagent type (species) or resource | Designation | Source or reference | Identifiers | Additional information |
|---|---|---|---|---|
| Cell line (*H. sapiens*) | A549 | ATCC | Cat # CCL-185 RRID:CVCL_0023 | |
| Cell line (*H. sapiens*) | A549$^{ACE2}$ | This study | N/A | |
| Cell line (*H. sapiens*) | HEK293FT | Thermo | Cat # R70007 RRID:CVCL_6911 | |
| Cell line (*H. sapiens*) | Huh-7.5 | tenOever lab | RRID:CVCL_7927 | |
| Cell line (*H. sapiens*) | Huh-7.5$^{ACE2}$ | This study | N/A | |
| Cell line (*H. sapiens*) | Caco-2 | ATCC | Cat # HTB-37 RRID:CVCL_0025 | |
| Cell line (*H. sapiens*) | Calu-3 | tenOever Lab | Cat # HTB-55 RRID:CVCL_YZ47 | |
| Antibody | Anti-GAPDH (rabbit monoclonal) | Cell Signalling | Cat # 2118 RRID: AB_561053 | (1:10,000), (1 µL) |
| Antibody | Anti-p24 (rabbit monoclonal) | Sino Biological | Cat # 11695-V08E RRID:N/A | (1:500), (10 µL) |
| Antibody | Anti-rhodopsin/C9 (mouse monoclonal) | Novus | Cat # NBP1-47602 RRID:AB_10010560 | (1:1000), (5 µL) |
| Antibody | IRDye 680RD donkey anti-rabbit | LI-COR | Cat # 926–68073; RRID:AB_10954442 | (1:5000), (1 µL) |
| Antibody | IRDye 800CW donkey anti-mouse | LI-COR | Cat # 926–32212; RRID: AB_621847 | (1:5000), (1 µL) |
| Antibody | Nucleocapsid (Human monoclonal) | Mount Sinai | Clone 1C7C7 RRID:N/A | Isolated by the Center for Therapeutic Antibody Discovery at the Icahn School of Medicine at Mount Sinai (1:1000), (1 µL) |
| Recombinant DNA reagent | pcDNA3.1-SARS2-Spike | Addgene | Cat # 145032 | |
| Recombinant DNA reagent | pcDNA3.1-SARS2-Spike-D614G | This study Addgene | Cat # 166850 | |
| Recombinant DNA reagent | pLentiEGFP | Addgene | Cat # 138152 | |
| Recombinant DNA reagent | pMD2.G | Addgene | Cat # 12259 | |
| Peptide, recombinant protein | Spike protein D614 | Sino Biological | Cat # 40591-V08H | |

*Continued on next page*

*Continued*

| Reagent type (species) or resource | Designation | Source or reference | Identifiers | Additional information |
|---|---|---|---|---|
| Peptide, recombinant protein | Spike protein G614 | Sino Biological | Cat # 40591-V08H3 | |
| Commercial assay or kit | psPAX2 | Addgene | Cat # 12260 | |
| Commercial assay or kit | Q5 site-directed mutagenesis kit | NEB | Cat # E0554S | |
| Commercial assay or kit | Luna Universal One-step qPCR kit | NEB | Cat # E3005L | |
| Software, algorithm | NextStrain | *Hadfield et al., 2018* | | https://nextstrain.org/ncov |
| Software, algorithm | FlowJo v10 | BD Bioscience | | |
| Software, algorithm | GraphPad Prism 8 | GraphPad | | |
| Software, algorithm | Image Lab 6.1 | BioRad | | |
| Software, algorithm | Image J 1.51 | NIH/LOCI | | |
| Software, algorithm | NetMHC 4.0 | *Andreatta and Nielsen, 2016* | | http://www.cbs.dtu.dk/services/NetMHC/ |
| Other | SARS-CoV-2 | CDC | Isolate USA-WA1/2020 (NR-52281) | Deposited by the Center for Disease Control and Prevention and obtained through BEI Resources, NIAID, NIH |

## SARS-CoV-2 genome analyses

For temporal tracking of D614G mutations in SARS-CoV-2 genomes, we used the Nextstrain analysis tool (https://nextstrain.org/ncov) with data obtained from GISAID (https://www.gisaid.org/) (*Hadfield et al., 2018*; *Shu and McCauley, 2017*). With the Nextstrain webtool, we visualized 3866 genomes using the 'clock' layout with sample coloring based on Spike 614 mutation status.

All complete SARS-CoV-2 genomes submitted before June 2, 2020 were obtained from GISAID. We excluded genomes classified by GISAID as low coverage and downloaded the remaining 23,755 high-coverage genomes. To classify each genome as D614 or G614, we flanked the mutation site with 11 nt of surrounding sequence context on each side and identified genomes matching either mutation. For 1652 genomes, we could not identify the mutation site and excluded these from further analysis. For the remaining 22,103 genomes, we were able to uniquely classify them as D614 or G614. Case-fatality rate data was downloaded on June 3, 2020 from the Johns Hopkins Coronavirus Resource Center (https://coronavirus.jhu.edu/data/mortality). For accurate estimation of D614G prevalence, we only included countries with at least nine genomes in GISAID.

## COVID-19 patient qPCR

Threshold cycle data and statistical test results for Sheffield quantitative PCR data from COVID-19 patients are from *Korber et al., 2020*. Threshold cycle data and statistical test results for University of Washington (UW) quantitative PCR data from COVID-19 patients are from *Wagner et al., 2020* (https://github.com/blab/ncov-D614G). For the Sheffield study, the reported threshold cycle was the median in each group. For the UW study, the reported threshold cycle was the mean in each group. The reported p-values were computed by the respective study authors using the Wilcoxon rank sum test.

## Cell culture

A549 (RRID:CVCL_0023) cells were obtained from ATCC, HEK293FT (RRID:CVCL_6911) cells were obtained from Thermo Scientific, and Huh-7.5 (RRID:CVCL_7927), Calu-3 (RRID:CVCL_YZ47), and Caco-2 (RRID:CVCL_0025) were a kind gift of B. tenOever (Mt. Sinai). A549, HEK293FT, Huh7.5, and Caco-2 were cultured in D10 media: Dulbecco's modified Eagle medium (DMEM) (Caisson Labs) supplemented with 10% Serum Plus II Medium Supplement (Sigma–Aldrich). Calu-3 were cultured in

Eagle's minimal essential medium (ThemoFisher) supplemented with 10% Serum Plus II Medium Supplement (Sigma–Aldrich). Cells were regularly passaged and tested for the presence of mycoplasma contamination (MycoAlert Plus Mycoplasma Detection Kit, Lonza).

## Spike plasmid cloning and lentiviral production

To express the D614 Spike, we used an existing cytomegalovirus (CMV)-driven SARS-CoV-2 plasmid (pcDNA3.1-SARS2-Spike, Addgene 145032) (**Shang et al., 2020**). To express the G614 Spike, we cloned pcDNA3.1-SARS2-SpikeD614G using the Q5 site-directed mutagenesis kit (NEB E0554S) and the following primers: 5′-CTGTACCAGGgCGTGAATTGCAC-3′ and 5′-CACGGCCACCTGGTTGCT-3′.

To make Spike-pseudotyped lentivirus, we co-transfected a d2EGFP-containing transfer plasmid (Addgene 138152) with accessory plasmid psPAX2 (Addgene 12260) and the pseudotyping plasmid (or omitted the pseudotyping plasmid to produce no-pseudotype lentivirus). Briefly, for each virus, a T-225 flask of 80% confluent HEK293T cells (Thermo) was transfected in OptiMEM (Thermo) using 25 µg of the transfer plasmid, 20 µg psPAX2, 22 µg Spike plasmid, and 175 µL of linear polyethylenimine (1 mg/mL) (Polysciences). Six hours post-transfection media was replaced with fresh D10 media, DMEM (Caisson Labs) with 10% Serum Plus II Medium Supplement (Sigma–Aldrich), with 1% bovine serum albumin (Sigma) added to improve virus stability. After 60 hr, viral supernatants were harvested and centrifuged at 3,000 rpm at 4°C for 10 min to pellet cell debris and filtered using 45 µm PVDF filters (CellTreat). The supernatant was then ultracentrifuged for 2 hr at 100,000 g (Sorvall Lynx 6000), and the pellet was resuspended overnight at 4°C in phosphate-buffered saline (PBS) with 1% bovine serum albumin (BSA).

## qPCR of Spike pseudoviruses

Viral RNA was isolated from 100 mL of 100× concentrated Spike D614 or G614 pseudotyped lentiviruses using 500 mL Trizol (Thermo 15596026) and following the Zymo Direct-zol RNA MicroPrep kit protocol. RNA was eluted with 15 mL RNase-free water. The RNA was then diluted 1:50 and 2 mL was used to perform a one-step qPCR protocol using Luna Universal One-step qPCR kit (NEB). Two primer sets were used: 5′-CGCTATGTGGATACGCTGC-3′ and 5′-GCGAAAGTCCCGGAAAGGAG-3′ that amplify WPRE and 5′-CGTGCAGCTCGCCGACCAC-3′ and 5′-CTTGTACAGCTCGTCCATGCC-3′ that amplify EGFP. qPCR was performed following the Luna Universal One-step qPCR kit protocol on a ViiA 384-well qPCR machine.

## Spike pseudovirus transductions

We plated 50,000 cells per well of a 48-well plate. The cells were transduced the following morning using the indicated pseudotyped lentiviral amounts plus media supplemented with polybrene 8 µg/mL to a final volume of 150 µL per well. The media was changed 8 hr post-transduction. The cells were analyzed by flow cytometry 72 hr post-transduction.

## ACE2 lentiviral cloning and ACE2 stable cell line overexpression

To generate pLenti-ACE2-Hygro (Addgene 161758), we amplified human ACE2 (hACE2) from pcDNA3.1-ACE2 (Addgene 1786) and cloned it into a lentiviral transfer pLEX vector carrying the hygromycin resistance gene using Gibson Assembly Master Mix (NEB E2611L). A 2A epitope tag was added to hACE2 at the C-terminus. Huh7.5$^{ACE2}$ and A549$^{ACE2}$ cell lines were generated by lentiviral transduction of ACE2. The protocol for lentiviral production was the same as above except we used the common lentiviral pseudotype (VSV-g) using plasmid pMD2.G (Addgene 12259). Transduced cells were selected with hygromycin at 50 µg/mL for Huh7.5$^{ACE2}$ and 500 µg/mL for A549$^{ACE2}$ for 10 days before use.

## Flow cytometry of transduced human cells

Cells were harvested and washed with Dulbecco's phosphate-buffered saline (Caisson Labs) twice. Cell acquisition and sorting was performed using a Sony SH800S cell sorter with a 100 µm sorting chip. We used the following gating strategy: (1) We excluded the cell debris based on the forward and reverse scatter; (2) doublets were excluded. For all samples, we recorded at least 5000 cells that pass the gating criteria described above. Gates to determine GFP+ cells were set based on control

GFP− cells, where the percent of GFP+ cells was set as <0.5% (background level). Flow cytometry analyses were performed using FloJo v10.

## ACE2 binding kinetics using bio-layer interferometry

The kinetics of the D614 and G614 Spike protein variants with hACE2 were analyzed using bio-layer interferometry on an Octet system (ForteBio, Octet RED96). Recombinant His-tagged and biotinylated human ACE2 protein (Sino Biological, Cat 10108-H08H-B) was immobilized on a Streptavidin (SA)-coated sensor. Loaded sensors were dipped into recombinant SARS-Cov-2 His-tagged Spike protein (D614 or D614G, Sino Biological, Cat # 40591-V08H and 40591-V08H3).

All proteins were diluted in kinetics buffer (0.1% wt/vol BSA, 0.02% Tween-20 in 1× PBS). Sensors were equilibrated in kinetics buffer for 10 min at room temperature preceding data acquisition, and experiments were performed at 30℃. Prior to ligand load, a baseline level was established for 60 s. hACE2 was loaded onto the sensor at 2.5 µg/mL for 180 s, followed by a sensor wash (180 s) and a second baseline establishment (60 s) in kinetics buffer. Analyte in concentrations ranging from 200 nM to 6.25 nM was associated for 240 s and dissociated for 240 s. To determine $K_D$ values for each variant, a reference sensor with loaded ligand, but no analyte was subtracted from the data before fitting. Data was fit using a 2:1 heterogeneous ligand model from association and dissociation rates. The analysis was carried out with Octet ForteBio Analysis 9.0 software. Using a 2:1 model, two $K_D$s were obtained. The higher affinity $K_D$s are consistent with previously published values, and the second $K_D$ values may represent a small amount of non-specific binding.

## Protein expression of ACE2 and Spike variants in human cells

HEK293FT cells were transiently transfected with equal amounts of Spike or ACE2 vectors using PEI. Cells were collected 18–24 hr post-transfection with TrypLE (Thermo), washed twice with PBS (Caisson Labs), and lysed with TNE buffer (10 mM Tris–HCl, pH 7.4, 150 mM NaCl, 1 mM EDTA, 1% Nonidet P-40) supplemented with protease inhibitor cocktail (Bimake B14001) for 1 hr on a rotator at 4℃. Cells lysates were spun for 10 min at 10,000 g at 4℃, and protein concentration was determined using the BCA assay (Thermo 23227). Whole-cell lysates (10 µg protein per sample) or 5 µL of 100× concentrated Spike-pseudotyped lentiviral particles were denatured in Tris–glycine sodium dodecyl sulfate (SDS) sample buffer (Thermo LC2676) and loaded on a Novex 4–12% Tris–glycine gel (Thermo XP04122BOX). PageRuler pre-stained protein ladder (Thermo 26616) was used to determine the protein size. The gel was run in 1× Tris–glycine–SDS buffer (IBI Scientific IBI01160) for about 120 min at 120 V. Protein transfer was performed using nitrocellulose membrane (BioRad 1620112) using prechilled 1× Tris–glycine transfer buffer (Fisher LC3675) with 20% methanol for 100 min at 100 V. Membranes were blocked with 5% skim milk dissolved in PBST (1× PBS + 1% Tween 20) at room temperature for 1 hr. Primary antibody incubations were performed overnight at 4℃ using the following antibodies: rabbit anti-GAPDH 14C10 (RRID: AB_561053) (0.1 µg/mL, Cell Signaling 2118S), rabbit p24 monoclonal antibody clone 002, which recognizes lentiviral capsid protein (1 µg/mL, Sino Biological 11695-V08E), and mouse anti-rhodopsin antibody clone 1D4 (RRID:AB_10010560) (1 µg/mL, Novus NBP1-47602), which recognizes the C9-tag added to the Spike proteins. Following the primary antibody, the blots were incubated with IRDye 680RD donkey anti-rabbit (AB_10954442) (0.2 µg/mL, LI-COR 926–68073) or with IRDye 800CW donkey anti-mouse (AB_621847) (0.2 µg/mL, LI-COR 926–32212) for 1 hr at room temperature. The blots were imaged using Odyssey CLx (LI-COR). Band intensity quantification was performed by first converting Odyssey multichannel TIFFs into 16-bit grayscale image (Fiji) and the then selecting lanes and bands in ImageLab 6.1 (BioRad). In ImageLab, background subtraction was applied uniformly across all lanes on the same gel.

## Epitope prediction using NetMHC

Since 9mer epitopes are most commonly presented by MHC receptors (*Sarkizova et al., 2020*), we constructed all possible 9mers surrounding the D/G 614 site in the Spike protein. We predicted binding affinities for five common HLA-A alleles and seven common HLA-B alleles using the NetMHC 4.0 prediction webserver (*Andreatta and Nielsen, 2016*) (http://www.cbs.dtu.dk/services/NetMHC/). For each peptide, we computed the difference in predicted affinity between the D614 and G614 variant using R/RStudio and visualized them using the pheatmap R package.

## SARS-CoV-2 *trans*-complementation assay

SARS-related coronavirus 2 (SARS-CoV-2), Isolate USA-WA1/2020 (NR-52281) was deposited by the Center for Disease Control and Prevention and obtained through BEI Resources, NIAID, NIH. SARS-CoV-2 was propagated in Vero E6 cells in DMEM supplemented with 2% FBS, 4.5 g/L D-glucose, 4 mM L-glutamine, 10 mM non-essential amino acids, 1 mM sodium pyruvate, and 10 mM HEPES.

For the trans-complementation assay, we co-transfected pcDNA3.1-SARS2-Spike or pcDNA3.1-SARS2-SpikeD614G (see above) with pcDNA3.1-ACE2 (Addgene 1786) at a 1:1 ratio (Spike: ACE2) using Lipofectamine 2000 (Thermo) as per the manufacturer's protocol. At specified time points (12, 18, or 24 hr post-infection), cells were fixed using 5% formaldehyde and immunostained for nucleocapsid (N) protein (clone 1C7C7, Center for Therapeutic Antibody Discovery at the Icahn School of Medicine at Mount Sinai) with DAPI to stain nuclei. Full wells were imaged and quantified for SARS-CoV-2-infected cells using a Celigo imaging cytometer (Nexcelom Biosciences). All infections with SARS-CoV-2 were performed with six biological replicates.

## Statistical analysis

Data analysis was performed using R/Rstudio 3.6.1 and GraphPad Prism 8 (GraphPad Software Inc). Specific statistical analysis methods are described in the figure legends where results are presented. Values were considered statistically significant for p-values below 0.05.

# Acknowledgements

We thank the entire Sanjana laboratory for support and advice. We are grateful to T Maniatis, M Legut, L Lu, D Ekiert, R Redler, K McGhee, C Lu, and M Prober for help with this work. ZD is supported by an American Heart Association postdoctoral fellowship (grant no. 20POST35220040). Postdoctoral fellowship support for TXJ is provided by the NIH (grant no. R01AI123155). GB is supported by PEW Biomedical Scholars (PEW-00033055), Searle Scholars Program (SSP-2018–2737) and NIH grant (grant no. R01AI147131). BRt is supported by the Marc Haas Foundation, the National Institutes of Health, and DARPA's PREPARE Program (HR0011-20-2-0040). NES is supported by New York University and New York Genome Center startup funds, National Institutes of Health (NIH)/National Human Genome Research Institute (grant no. R00HG008171, DP2HG010099), NIH/National Cancer Institute (grant no. R01CA218668), Defense Advanced Research Projects Agency (grant no. D18AP00053), the Sidney Kimmel Foundation, the Melanoma Research Alliance, and the Brain and Behavior Foundation.

# Additional information

### Competing interests

Neville E Sanjana: N.E.S. is an advisor to Vertex. The other authors declare that no competing interests exist.

### Funding

| Funder | Grant reference number | Author |
|---|---|---|
| American Heart Association | 20POST35220040 | Zharko Daniloski |
| National Institute of Allergy and Infectious Diseases | R01AI123155 | Tristan X Jordan |
| Pew Charitable Trusts | PEW-00033055 | Gira Bhabha |
| Searle Scholars Program | SSP-2018-2737 | Gira Bhabha |
| National Institute of Allergy and Infectious Diseases | R01AI147131 | Gira Bhabha |
| Defense Advanced Research Projects Agency | HR0011-20-2-0040 | Benjamin R tenOever |
| National Human Genome Research Institute | DP2HG010099 | Neville E Sanjana |

| National Cancer Institute | R01CA218668 | Neville E Sanjana |
|---|---|---|
| Defense Advanced Research Projects Agency | D18AP00053 | Neville E Sanjana |
| Sidney Kimmel Foundation | | Neville E Sanjana |
| Melanoma Research Alliance | | Neville E Sanjana |
| Brain and Behavior Research Foundation | | Neville E Sanjana |
| NIH | R00HG008171 | Gira Bhabha |

The funders had no role in study design, data collection and interpretation, or the decision to submit the work for publication.

## Author contributions

Zharko Daniloski, Conceptualization, Formal analysis, Validation, Investigation, Visualization, Methodology, Writing - original draft, Project administration, Writing - review and editing; Tristan X Jordan, Juliana K Ilmain, Formal analysis, Investigation, Visualization, Methodology, Writing - review and editing; Xinyi Guo, Data curation, Formal analysis, Investigation, Visualization, Methodology, Writing - review and editing; Gira Bhabha, Resources, Formal analysis, Supervision, Funding acquisition, Methodology, Project administration, Writing - review and editing; Benjamin R tenOever, Resources, Formal analysis, Supervision, Funding acquisition, Investigation, Methodology, Project administration, Writing - review and editing; Neville E Sanjana, Conceptualization, Resources, Data curation, Formal analysis, Supervision, Funding acquisition, Investigation, Visualization, Methodology, Writing - original draft, Project administration, Writing - review and editing

## Author ORCIDs

Zharko Daniloski https://orcid.org/0000-0002-3453-0849
Tristan X Jordan https://orcid.org/0000-0002-0602-2871
Juliana K Ilmain http://orcid.org/0000-0002-9507-5069
Gira Bhabha http://orcid.org/0000-0003-0624-6178
Benjamin R tenOever https://orcid.org/0000-0003-0324-3078
Neville E Sanjana https://orcid.org/0000-0002-1504-0027

## Decision letter and Author response

Decision letter https://doi.org/10.7554/eLife.65365.sa1
Author response https://doi.org/10.7554/eLife.65365.sa2

# Additional files

## Supplementary files

• Transparent reporting form

## Data availability

All data generated or analyzed in this study are included in this published article and its supplementary information files. The Spike D614G expression plasmid has been deposited to Addgene (#166850).

The following previously published datasets were used:

| Author(s) | Year | Dataset title | Dataset URL | Database and Identifier |
|---|---|---|---|---|
| Korber B | 2020 | Spike mutation pipeline reveals the emergence of a more transmissible form of SARS-CoV-2 | https://doi.org/10.1101/2020.04.29.069054 | bioRxiv, 10.1101/2020.04.29.069054 |
| Wagner C | 2020 | Comparing viral load and clinical outcomes in Washington State across D614G mutation in spike | https://github.com/blab/ncov-wa-d614g/blob/39f87ba09a8e9a- | GitHub, README.md |

| protein of SARS-CoV-2 | d1789052039f8602-a445364b9e/README.md |
|---|---|

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
