## [Decision Letter]

**Acceptance summary:**

In agreement with the accumulating evidence that the D614G mutation in the SARS-CoV-2 Spike protein is associated with a selective advantage for viral spread, the present study shows that this mutation increases transduction by Spike containing lentiviral particles as well as the infectiousness of genuine SARS-CoV-2 in human lung epithelial cells. The authors further report that the D614G changes hardly affect ACE2 receptor binding, but do affect the sensitivity of the Spike protein to proteolytic cleavage, which might help to explain effects on viral infectivity.

**Decision letter after peer review:**

Thank you for submitting your article "The Spike D614G mutation increases SARS-CoV-2 infection of multiple human cell types" for consideration by *eLife*. Your article has been reviewed by three peer reviewers, including Frank Kirchhoff as the Reviewing Editor and Reviewer #1, and the evaluation has been overseen by Sara Sawyer as the Senior Editor.

The reviewers have discussed the reviews with one another and the Reviewing Editor has drafted this decision to help you prepare a revised submission.

Summary:

Daniloski and colleagues examined the functional consequences of a D614G mutation in the SARS-CoV-2 Spike protein that seems to be associated with a selective advantage for viral spread in the human population. They confirm that viruses harboring mutation D614G became dominant in the pandemic and further show that this mutation increases transduction by Spike containing lentiviral particles in several cell lines including human lung epithelial cells. The authors further report that the D614G change hardly affects ACE2 receptor binding but renders the Spike protein less sensitive to proteolytic cleavage. Finally, they conclude that their results are important for the efficacy of Spike-based vaccines.

Determining the impact of D614G variants on the course of the pandemic and clarifying how D614G modulates virus-host cell interactions is important. Recently, several studies have been published on this subject that show, among other findings, that D614G increases entry and viral fitness. Moreover, mechanistic explanations for increased entry have been provided, including G614 spike being more frequently in an ACE2 binding-competent conformation (PMID: 33106671, PMID: 32730807, PMID: 32991842, PMID: 33184236, PMID: 33243994). In the light of these studies, the novelty of the findings reported in the present study is limited but the results are still of significant interest. For most part the data seem solid. As outlined below, however, some overstatements should be avoided and conclusions on the mechanism seem rather preliminary based on the results presented. Furthermore, limitations of the present study (e.g. that all but one cell line overexpress ACE2) should be clearly stated and discrepancies to other studies investigating the effect on the D614G change need to be discussed.

Essential revisions:

1) "This result has important implications for the efficacy of Spike-based vaccines currently under development in protecting against this recent and highly-prevalent SARS-CoV-2 isolate." The authors did not analyze whether the D614G mutation affects vaccine efficiency and recent data suggest that this might not be the case. Thus, this statement should be removed from the Abstract.

2) Enhancing effects varied from about 2-to 8-fold and the strongest effects were obtained in cells overexpressing ACE2. The authors should not only mention the maximum effect but provide the range in the Abstract. More importantly, transduction data can hardly be evaluated in the absence of data on S protein incorporation into particles. Increased entry driven by D614G spike might simply reflect increased particle incorporation in the pseudotyping system used by the authors. This should be examined. In addition, the significance of the results would be strengthened by analyses of D614G spike-mediated transduction of human lung cell lines or (better) primary lung cells that endogenously express ACE2 and TMPRSS2. Otherwise the caveat that the strongest effects were observed under conditions of ACE2 overexpression should already be clearly stated in the Abstract.

3) Cleavage was only analyzed in HEK293T cells and only total spike present in cells is shown. However, the cleavage status of spike present in particles (and not in cells) impacts spike protein-mediated entry. This could easily be tested; however, Ozono and colleagues reported that the effect of D614G is independent of virion incorporation of the S protein. Unlike the present study, Yurkovetskiy and colleagues reported that D614G affinity for ACE2 is reduced due to a faster dissociation rate and He et al. reported that the G614 S protein was cleaved more efficiently by serine protease elastase-2. These discrepancies should be discussed.

4) The observation that D614 increases SARS-CoV-2 infection of human cells grown in culture is meanwhile well established. All relevant literature, e.g. https://www.cell.com/cell/fulltext/S0092-8674(20)30877-1; https://www.nature.com/articles/s41586-020-2895-3; https://www.nature.com/articles/s41392-020-00392-4 should be cited and priority claims avoided. The authors should discuss the different models presented in the literature regarding exactly how D614G increases infectivity, and they should compare and contrast their results with these other hypotheses. There are several other locations where text in this manuscript needs to be updated. For instance, there are many more than 10 vaccines in clinical trials at the moment (Results).

---

## [Author Response]

Essential revisions:1) "This result has important implications for the efficacy of Spike-based vaccines currently under development in protecting against this recent and highly-prevalent SARS-CoV-2 isolate." The authors did not analyze whether the D614G mutation affects vaccine efficiency and recent data suggest that this might not be the case. Thus, this statement should be removed from the Abstract.

Since our initial work, we have indeed learned much more about this mutation and vaccine efficiency. We agree with the reviewers and have removed this statement from the Abstract.

2) Enhancing effects varied from about 2-to 8-fold and the strongest effects were obtained in cells overexpressing ACE2. The authors should not only mention the maximum effect but provide the range in the Abstract. More importantly, transduction data can hardly be evaluated in the absence of data on S protein incorporation into particles. Increased entry driven by D614G spike might simply reflect increased particle incorporation in the pseudotyping system used by the authors. This should be examined. In addition, the significance of the results would be strengthened by analyses of D614G spike-mediated transduction of human lung cell lines or (better) primary lung cells that endogenously express ACE2 and TMPRSS2. Otherwise the caveat that the strongest effects were observed under conditions of ACE2 overexpression should already be clearly stated in the Abstract.

We thank the reviewers for these helpful comments and address each point below:

a) In the Abstract and text of the revised manuscript we replaced the maximal fold-change with the range. We additionally provide separate ranges for cell lines with endogenous ACE2 expression and those that overexpress ACE2, which highlights the stronger effects under conditions of ACE2 overexpression. The Abstract now reads:

“The increased transduction with spike D614G ranged from 1.3 to 2.4-fold in Caco-2 and Calu-3 cells expressing endogenous ACE2, and 1.5 to 7.7-fold in A549-ACE2 and Huh7.5-ACE2 overexpressing ACE2.”

b) Instead of the maximal fold-change, we now display the full range of the transduction fold-change (G614/D614) in the main figures (Figure 2D).

c) As suggested, we measured the Spike incorporation into pseudotyped lentiviral particles by western blot. We quantified the total Spike (uncleaved and cleaved) and normalized it to the lentiviral capsid protein p24. We observed no significant difference in Spike incorporation between lentiviral particles pseudotyped with Spike D614 or G614 (Figure 3—figure supplement 1).

d) As recommended by the reviewers, we measured the transduction efficiency of both variants in a human lung cell line that endogenously expresses ACE2 (Calu-3). Consistent with the other cell line with endogenous ACE2 expression (Caco-2), we observed that viral particles pseudotyped with Spike G614 were about 2-fold more efficient at transducing Calu-3 cells compared to D614 variants.

3) Cleavage was only analyzed in HEK293T cells and only total spike present in cells is shown. However, the cleavage status of spike present in particles (and not in cells) impacts spike protein-mediated entry. This could easily be tested; however, Ozono and colleagues reported that the effect of D614G is independent of virion incorporation of the S protein. Unlike the present study, Yurkovetskiy and colleagues reported that D614G affinity for ACE2 is reduced due to a faster dissociation rate and He et al. reported that the G614 S protein was cleaved more efficiently by serine protease elastase-2. These discrepancies should be discussed.

We thank the reviewers for this comment. In our revised manuscript, we quantified Spike cleavage (and total Spike incorporated) in pseudotyped lentiviral particles by western blot. We observe a small but significant reduction of Spike cleavage in G614 compared to D614, consistent with our HEK293 transfection experiments (Figure 3G-I).

In addition, in the revised manuscript we have added a discussion of our results in light of work by Ozono, Yurkovetskiy, and He:

“We also found that G614 Spike is more resistant to proteolytic cleavage during production of the protein in host cells as well as on pseudotyped lentiviral particles, suggesting that replicated virus produced in human cells may be more infectious due to a greater proportion of functional (uncleaved with receptor-binding domain) Spike protein per virion. […] Yurkovetskiy and colleagues reported that, with purified Spike full-length trimers, the G614 mutation shifts the Spike protein conformation to an ACE2-binding competent state(Yurkovetskiy et al., 2020), suggesting that Spike full length trimers behave differently than the Spike S1 monomers that we investigated in this study.”

4) The observation that D614 increases SARS-CoV-2 infection of human cells grown in culture is meanwhile well established. All relevant literature, e.g. https://www.cell.com/cell/fulltext/S0092-8674(20)30877-1; https://www.nature.com/articles/s41586-020-2895-3; https://www.nature.com/articles/s41392-020-00392-4 should be cited and priority claims avoided. The authors should discuss the different models presented in the literature regarding exactly how D614G increases infectivity, and they should compare and contrast their results with these other hypotheses. There are several other locations where text in this manuscript needs to be updated. For instance, there are many more than 10 vaccines in clinical trials at the moment (Results).

Thank you for this comment. We have cited all relevant D614G published work, eliminated priority claims and also updated other numbers throughout the manuscript. In addition, we have updated the manuscript in various locations, including the vaccine number in clinical trials.

We also included a paragraph in the Discussion that explains the models used to study D614G and the mechanisms explored by various studies:

“Several other groups have also reported that the D614G results in increased viral fitness and infection efficiency(Hu et al., 2020; Jiang et al., 2020; Li et al., 2020; Ozono et al., 2020; Plante et al., 2020; Yurkovetskiy et al., 2020; Zhang et al., 2020). […] Altogether, these differences illustrate the advantages and limitations of using pseudoviruses and isogenic viruses that should be taken into consideration in future studies.”